# Heterogeneity of genetic sequence within quasi-species of influenza virus revealed by single-molecule sequencing

Kenji Tamao[1], Hiroyuki Noji[1,2], Kazuhito Tabata[1,2]*

[1]Department of Applied Chemistry, Graduate School of Engineering, The University of Tokyo, Tokyo, Japan; [2]Research Institute of Planetary Health, The University of Tokyo, Tokyo, Japan

## eLife Assessment

This study is an **important** contribution to the field of viral sequencing, providing methods for more accurate characterization of viral genetic diversity using long-read sequencing and unique molecular identifiers (UMIs). Although it is a small pilot study, it shows promise as a **convincing**, validated methodology with broad applicability.

**\*For correspondence:**
tabatak@g.ecc.u-tokyo.ac.jp

**Competing interest:** The authors declare that no competing interests exist.

**Abstract** Influenza viruses exhibit high mutation rates and extensive genetic diversity, which hinder effective vaccine development and facilitate immune evasion (Taubenberger and Morens, 2006; Barr et al., 2010). These mutations arise from the error-prone viral RNA-dependent RNA polymerase, generating highly heterogeneous viral populations within individual hosts that conform to the quasi-species model of a cloud of related genomes evolving under selection (Domingo et al., 2012). Accurate characterization of this intra-host diversity is crucial for understanding viral evolution and improving vaccine design, yet conventional RNA sequencing often fails to detect low-frequency variants because of technical errors during sample preparation and sequencing. Here, we implement a single unique molecular identifier strategy that reduces sequencing artifacts and achieves an error rate of $\sim 10^{-5}$, enabling single-particle–level quantification of quasi-species diversity. Mutation frequencies greatly exceeding background error confirm their biological origin, while information-theoretic metrics such as Shannon entropy and Jensen–Shannon divergence reveal non-random mutation distributions under selective constraints. This framework supports detailed studies of intra-host viral evolution and may inform artificial intelligence-driven prediction of mutational trajectories and more effective influenza vaccine strategies.

## Introduction

Influenza is a seasonal illness caused by the influenza virus (*Cox et al., 1994*; *Barr et al., 2010*; *Ampofo et al., 2012*). The influenza virus has been observed to exhibit a high mutation rate and antigenic diversity, which has the potential to trigger pandemics and exert significant social and economic impacts (*Taubenberger and Morens, 2006*; *Pardi et al., 2018*).

In the context of influenza, vaccine development has been a prominent preventive strategy. However, the rapid mutation of the virus can lead to fluctuations in vaccine efficacy on an annual basis, thereby complicating precise prediction and comprehensive prevention of infection difficult (*Pardi et al., 2018*; *Belongia et al., 2016*).

Mutations in influenza viruses are primarily introduced by RNA-dependent RNA polymerase (RdRp) (*Zhu et al., 2009*). The high error rate characteristic of RdRp enables the virus to undergo rapid

evolution, thereby acquiring the capacity to evade existing immune responses (*Parvin et al., 1986*). It has been suggested that mutant strains that evade antibody-mediated immunity while retaining infectivity tend to exhibit specific patterns (*Visher et al., 2016*; *Skountzou et al., 2007*). Further research is necessary to elucidate these predictable mutation pathways (*Koel et al., 2013*). Recently, bioinformatics analyses that employ artificial intelligence (AI) have played a key role in investigating viral evolution mechanisms and exploring strategies for future vaccine design (*Deznabi et al., 2020*; *Thadani et al., 2023*; *Yan and Zhao, 2023*; *Shah et al., 2024*). However, enhancing the precision of AI-driven mutation prediction necessitates the consolidation of extensive global data. This effort is hampered by spatial and temporal limitations, which restrict the amount of data that can be collected and act as a bottleneck for enhancing the accuracy of predictions (*Dellicour et al., 2021*).

Currently, the genome sequences that are registered in viral genome databases represent the mean sequences of viral particle populations. However, due to the error-prone nature of viral RNA polymerase, viral populations exhibit high genetic diversity within a single host (*Domingo et al., 2012*; *Peck and Lauring, 2018*). This genetic diversity within viral populations can be described by the quasi-species model, which envisions a cloud of related viral genomes centered around a dominant sequence (*Sanjuán et al., 2004*). Quasi-species formation plays a crucial role in viral survival strategies, allowing the population to rapidly adapt to environmental pressures such as host immunity or antiviral drugs through collective evolution (*Bull et al., 2008*). The development of technology capable of accurately measuring the distribution of genome sequences within a virus population would facilitate a comprehensive evaluation of the types and frequencies of such precursor variants. It is anticipated that these data will play a significant role in the field of AI-based mutation prediction. Accordingly, the quasi-species concept serves as a fundamental framework for understanding viral evolution (*Wang et al., 2021*). The accurate quantification of quasi-species diversity is pivotal in facilitating a more profound comprehension of mutation dynamics, selective pressures, and evolutionary potential within a viral population (*Sardanyés et al., 2024*)—crucial not only for anticipating future dominant variants but also for detecting low-frequency mutations associated with drug resistance or antigenic shifts.

In current RNA sequencing technology, is plagued by the presence of errors that are introduced during reverse transcription process. The error rate has been reported to be $10^{-4}$ to $10^{-5}$ per nucleotide when using SuperScript IV reverse transcriptase, as utilized in the present study (*Zucha et al., 2020*; *Martín-Alonso et al., 2021*). Accordingly, a target sequencing error rate of $10^{-5}$ was established, and the single unique molecular identifier (sUMI) method was employed. This method has been shown to minimize sample loss and streamline procedures, thereby facilitating the quantification of genomic diversity within influenza virus particle populations (*Westfall et al., 2023*).

In this study, we demonstrated that the sUMI method can effectively reduce sequencing error rates to the $10^{-5}$ level. This method involves the ligation of UMIs directly to DNA amplified by cloning, followed by PCR and sequencing. Subsequently, an analysis of quasi-species was conducted on a population of influenza virus particles that had proliferated from a single particle and quantified the variation in their genome sequences was quantified (*Figure 1*). Comparison of this variation with the mutation rate of RNA synthesized by in vitro transcription showed that viral populations harbored significantly more mutations, indicating they arose during viral replication. Furthermore, we employed information-theoretic metrics, including Jensen–Shannon divergence (JSD) (*Menéndez et al., 1997*) and Shannon entropy (*Shannon, 1948*), to analyze the distribution of mutations. Our analysis revealed that specific positions within the viral genome demonstrate selective propagation of mutations.

These results provide a novel approach for high-accuracy detection of minor genetic mutations within influenza virus populations and offer valuable insights for understanding viral evolution and designing future vaccines.

## Results
### Confirmation of reduction of error rate by sequencing UMI-ligated DNA

Initial findings indicate that establishing a threshold for UMI redundancy leads to a substantial reduction in sequencing errors attributable to the methodology. A single plasmid molecule was amplified via *Escherichia coli* cloning and subsequently linearized by restriction enzyme digestion. An oligonucleotide containing a primer-binding site and a UMI sequence at its 5' end was then ligated to the

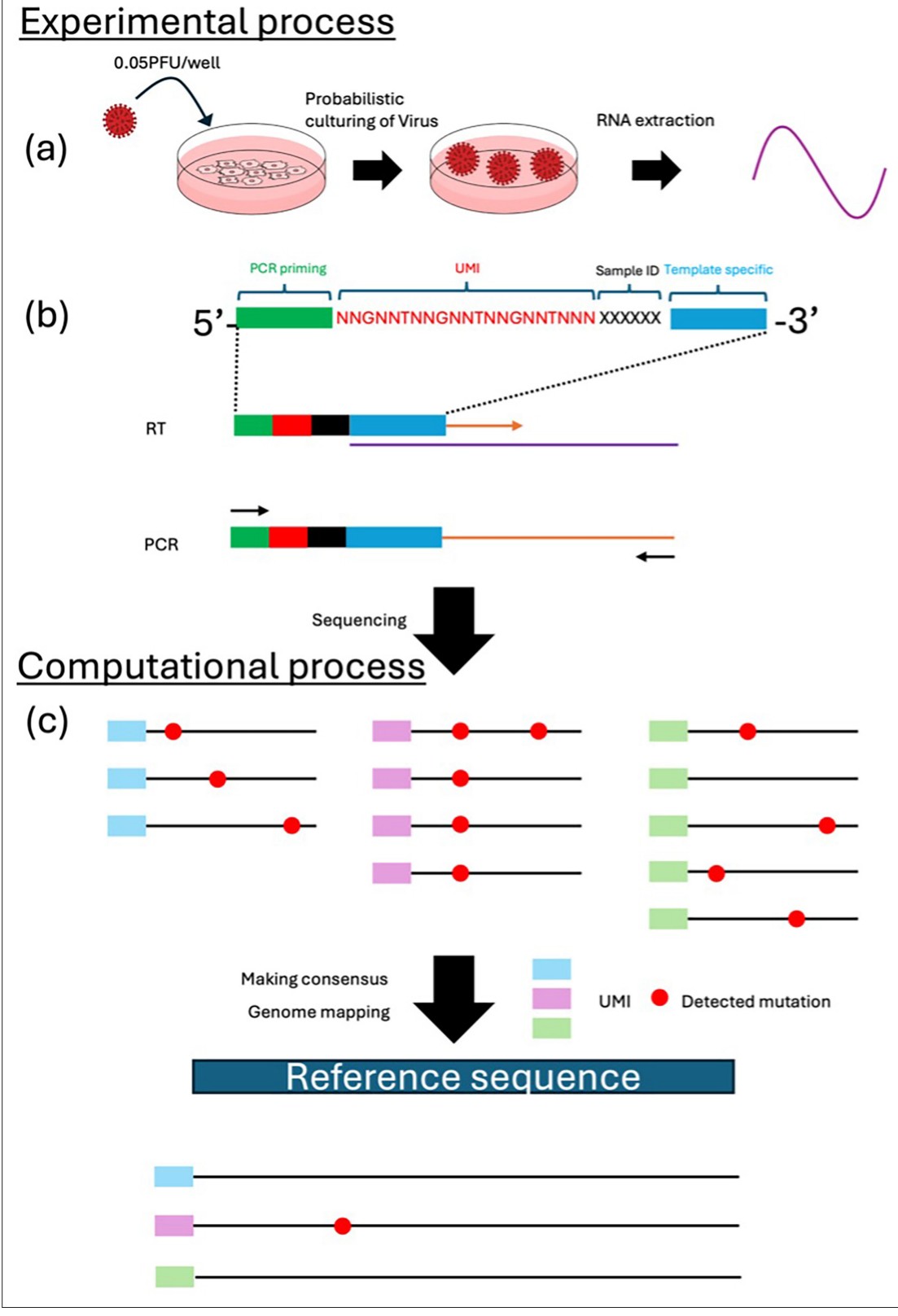

**Figure 1.** Overview of the process of obtaining genomic diversity of influenza virus in a population replicated from a single particle. (**a**) Madin–Darby Canine Kidney cells were infected with influenza virus at a concentration of 0.05 PFU/well. After 3–4 days, supernatants were collected from wells where cellular tissue destruction had occurred, and viral RNA was extracted. (**b**) The RT primer consisted of a 25-nucleotide PCR priming site at the 5' end, followed by a 6-nucleotide sample ID, a 21-nucleotide unique molecular identifier (UMI) containing a 15-nucleotide random nucleotide, and a

*Figure 1 continued on next page*

*Figure 1 continued*

12-nucleotide sequence complementary to the template. cDNA was synthesized via reverse transcription, followed by PCR amplification using primers containing the PCR priming site sequence and gene-specific sequence. (**c**) The sequences were grouped according to their UMI. For each unique UMI, the original RNA sequence was reconstructed by majority voting among identical sequences. The reconstructed RNA molecules were mapped to the reference genome, and the distribution of mutations was quantified.

linearized plasmid. The resulting UMI-tagged linear plasmid was amplified by PCR and subjected to sequencing using the Revio system. The resulting single-molecule sequences were then subjected to analysis using the umierrorcorrect function. During the analysis, we applied GroupSize thresholds of 1, 2, and 3 were applied, and only reads that met each threshold were used to generate consensus sequences for identical UMIs. These consensus sequences were then mapped to the reference sequence. Consequently, the error rates for GroupSize thresholds of 1, 2, and 3 were $3.59×10^{-4}$, $1.35×10^{-4}$, and $3.81×10^{-5}$, respectively (*Figure 2a*). When the threshold was set to 4, no errors were observed among 37,307 base pairs. These findings demonstrate that filtering sequences based on UMI redundancy can significantly reduce method-derived errors, improving sequencing accuracy by nearly an order of magnitude.

It is widely acknowledged that PacBio sequencers exhibit elevated error rates in homopolymer regions, defined as stretches of identical nucleotides (*Daly et al., 2021*; *Pourmohammadi et al., 2023*). In this study, the term 'homopolymer region' was defined as a sequence containing three or more identical consecutive bases, including one base on each side. Subsequently, an analysis was conducted to compare the error rates observed in homopolymer and non-homopolymer regions. In accordance with preceding reports, elevated error rates were detected in homopolymer regions. In the sequencing data of UMI-tagged linear plasmids employing a GroupSize threshold of 3, the error rate in homopolymer regions was $7.11×10^{-5}$, whereas the error rate in non-homopolymer regions was $1.66×10^{-5}$ (*Figure 2b*). As the GroupSize threshold increased from 1 to 3, a decrease in the discrepancy between homopolymer and non-homopolymer regions was observed. This finding indicates that the averaging of sequences with identical UMIs effectively mitigates sequencing errors, particularly in homopolymer regions.

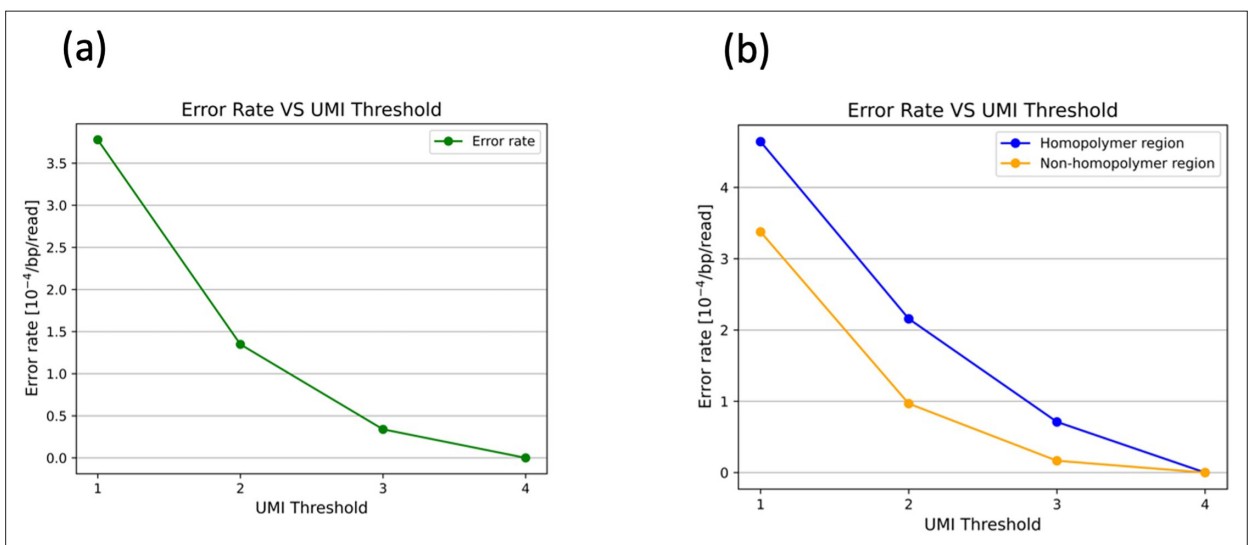

**Figure 2.** Error rates of unique molecular identifier (UMI)-ligated plasmid filtered by different group size. The plasmid, replicated by cloning in *E. coli*, was linearized using restriction enzyme treatment. A UMI was then attached through ligation, followed by amplification via PCR and subsequent sequencing. After sequencing, consensus sequences were obtained from reads sharing the same UMI, allowing for the reconstruction of single-molecule sequences. (**a**) Changes in the error rate of linearized plasmids when filtered based on group size (UMI duplication detection count). (**b**) Changes in the error rate in homopolymer and non-homopolymer regions when filtered based on group size.

## Quantification of error rate caused by RT and comparison with virus-derived mutation

In order to ascertain the mutation distribution of viral populations derived from a single virus particle, we sequenced viral genomic RNA extracted from infected cells and subjected it to sequencing. In order to differentiate between biological mutations and technical errors, a comparison was made with in vitro transcribed RNA derived from plasmids.

Madin–Darby Canine Kidney (MDCK) cells were seeded in a 6-well plate and infected with influenza virus at a multiplicity of infection (MOI) of 0.05 PFU/well. Following a 72-hour incubation period, wells exhibiting cytopathic effects (CPEs) were harvested. Of the 96 wells, 4 exhibited clear CPE, suggesting successful viral amplification from single particles. The supernatants from these wells were collected as independent viral populations, and total viral genomic RNA was extracted. RT-PCR analysis of the NA gene confirmed that virus replication occurred only in samples where CPE was observed (*Figure 3—figure supplement 1*).

For each viral RNA sample, reverse transcription (RT) was performed with the ligation of UMIs, followed by PCR amplification and sequencing. To control for errors introduced during RT-PCR and sequencing, we prepared plasmids containing sequences homologous to the PB2 and HA segments under a T7 promoter and synthesized RNA via in vitro transcription. These in vitro transcripts were subsequently processed using the same UMI-based RT-PCR and sequencing protocol. Distribution of the number of consensus sequences after filtering based on UMI group size for each gene and population is shown in *Figure 3—figure supplement 2*.

In order to differentiate between biological mutations and technical errors, a method was developed that involved mapping viral reads to the consensus sequence obtained from each viral population and the in vitro transcribed reads to the reference plasmid sequence that was utilized for transcription. The observed error rates in the in vitro transcripts are considered to reflect technical errors associated with RT and sequencing. Conversely, the higher error rates observed in the viral RNA are attributed to a combination of biological mutations and technical noise.

It is noteworthy that viral RNA exhibited a consistent pattern of elevated error rates (*Figure 3a*). For the PB2 segment, the error rate for viral RNA was $1.19 \times 10^{-4}$, whereas that for in vitro transcribed PB2-like RNA was $2.22 \times 10^{-6}$. Accordingly, the probability that a detected mutation in PB2 originates from viral replication rather than experimental error was calculated as $1 - (2.22 \times 10^{-6})/(1.19 \times 10^{-4}) \approx$ 98.1%. Segment-specific error rates ranged from $1.17 \times 10^{-4}$ to $2.19 \times 10^{-4}$ (*Supplementary file 1*), and thus the PB2 value is within the genome-wide range. These error rates were significantly higher than those of in vitro transcribed PB2-like RNA, indicating that detected mutations are highly likely to originate from viral replication.

These results demonstrate that the majority of detected mutations are derived from biological processes during viral replication rather than technical artifacts. Consequently, this approach facilitates single-molecule genome sequencing and precise profiling of mutation heterogeneity within viral populations.

## Mutation profile of RNA extracted from a virus population derived from a single virus clone

*Supplementary file 1* presents a summary of the mutation frequency, transition/transversion ratio (Ti/Tv), and the ratio of nonsynonymous to synonymous substitutions (dN/dS) for each gene. The mutation detection rate across coding regions is illustrated in *Figure 3b*. The observed Ti/Tv ratios ranged from 2 to 5, which is consistent with previously reported values (*Pauly et al., 2017*; *Duchêne et al., 2015*). In the absence of any specific selection pressure applied within the cell culture system, the dN/dS values were found to be generally below 1, thereby suggesting purifying selection as a mechanism for maintaining protein function. It is noteworthy that the NA gene exhibited dN/dS values approaching or exceeding 1, indicating a heightened degree of evolutionary plasticity.

An investigation was conducted into the presence of defective viral genomes that exhibited partial deletions. This investigation was motivated by the findings of prior studies, which reported the existence of internally deleted genomes in influenza viruses (*Alnaji et al., 2021*; *Penn et al., 2022*). Among the 546,580 molecules that were analyzed, 837 (0.15%) contained continuous deletions longer than 10 nucleotides. However, no such deletions were identified in the in vitro transcribed RNAs, suggesting that these deletions likely emerged during the viral replication process (*Figure 3—figure*

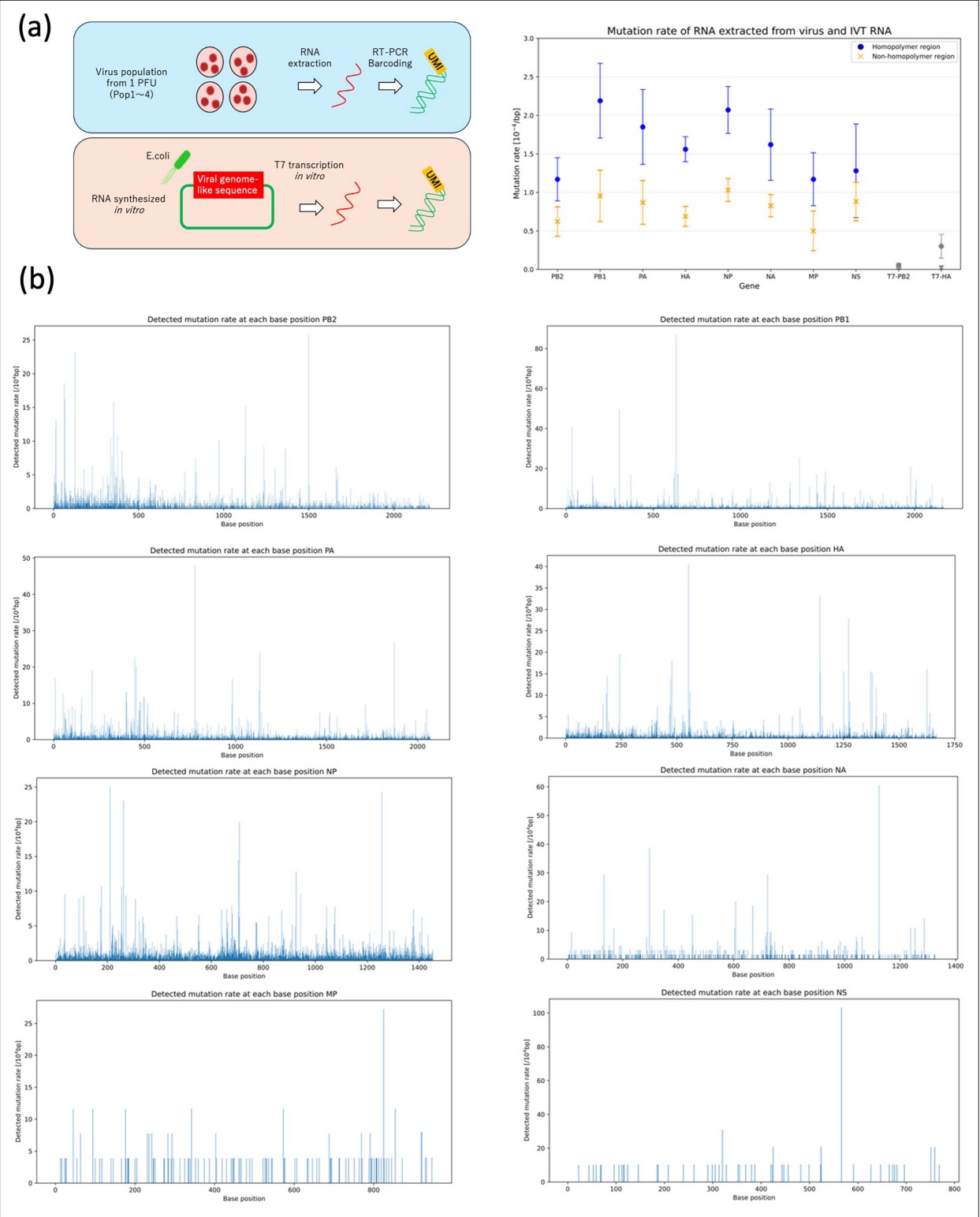

**Figure 3.** Error rates obtained from virus extracted RNA and in vitro transcribed RNA. (**a**) Sequencing results of RNA extracted from a virus population propagated from a single molecule and RNA synthesized by in vitro transcription from a plasmid. The viral RNA represents the average mutation rate obtained from four virus populations, with bars indicating the standard deviation (n=4). The in vitro transcribed RNA represents the average of two experiments, with bars indicating the standard deviation (n=2). (**b**) The bar graph of mutation rates obtained at each position within the coding region of the extracted viral RNA. The horizontal axis represents the nucleotide position within the coding region.

The online version of this article includes the following source data and figure supplement(s) for figure 3:

*Figure 3 continued on next page*

*Figure 3 continued*

**Source data 1.** All mutations list in amino acid form.

**Source data 2.** All mutations list in nucleotide form.

**Figure supplement 1.** Population formation from a single PFU.

**Figure supplement 2.** Distribution of the number of consensus sequences after filtering based on unique molecular identifier (UMI) group size for each gene and population.

**Figure supplement 3.** The proportion of sequences with ≧ 10 nt deletions.

*supplement 3*). Conversely, a mere 13 molecules (0.0024%) exhibited insertions that spanned more than 100 nucleotides. It should be noted that the RT-PCR primer design strategy did not take into account sequences with consecutive deletions at the termini, but only sequences with consecutive deletions in the center of the genome.

On average, over 10,000 consensus reads were obtained per gene after UMI-based error correction (group size ≥3). Independent digital particle counting estimated that each viral culture contained approximately $\sim 10^{13}$ viral particles. Under these conditions, variants present at a frequency of at least ≥0.1% in the population can be detected with >95% probability, exceeding the background error rate shown in *Figure 3a*.

In order to assess the applicability of this approach for comprehensive detection of drug-resistant mutations, a model was developed to demonstrate how mutant frequencies change with fitness using a competitive logistic growth model. This model is predicated on the premise that host cell resources are limited, and that there is a transition from exponential growth to saturation. It is widely employed for the estimation of effective reproduction numbers (*Hedge et al., 2013*; *Skums et al., 2018*; *Bing et al., 2020*; *Kartono et al., 2021*). The minimum relative fitness (compared to wild type) necessary for mutants to surpass this threshold, depending on their initial abundance, is shown in *Figure 4*. Even late-arising mutants (1 in $10^{11}$) with ≥5-fold fitness advantage could reach 0.1% of the population, indicating that such adaptive mutations can be detected by this method.

## Characteristics of the observed mutation distribution and deviation from the Poisson distribution

In order to investigate the characteristics of the mutation distribution in viral populations, a comparison was made between the distribution of genome sequences derived from viral RNA and that of RNA synthesized by in vitro transcription. For each specific nucleotide position, we constructed the distribution $P$ was constructed by defining $P(k)$ as the number of positions at which mutations were observed k times. Assuming that mutations introduced by T7 RNA polymerase, reverse transcriptase, and PCR enzymes occur randomly, the sequencing errors detected in in vitro transcribed RNA are expected to approximate a Poisson distribution. Conversely, if mutations introduced during viral replication are entirely random with respect to genomic location, then the mutation distribution of viral RNA should also adhere to a Poisson distribution. Conversely, substantial deviations from a Poisson distribution may indicate the presence of selection pressures, such as differences in replication efficiency associated with viral protein function or genomic sequence features.

The quantification of the discrepancy between the experimentally observed mutation count distribution observed through experimentation and the corresponding Poisson distribution with a mean count ($\lambda$) was achieved through the implementation of JSD. The JSD is a symmetric and normalized measure of relative entropy that is both symmetric and normalized. It takes on values approaching 1, indicating greater dissimilarity between distributions. The results indicated that the entire gene set, the mutation distributions of viral RNA exhibited a statistically significant deviation from the Poisson distribution when compared with those of in vitro transcribed RNA (*Figure 4a*). These findings imply that the observed mutation patterns in viral populations are not simply the result of neutral, randomly introduced mutations but reflect constraints imposed by selection related to protein function or replication efficiency. However, it is noteworthy that many minor mutations are introduced in a manner that approximately follows a Poisson distribution. These results clearly demonstrate that most mutations are introduced randomly, contributing to the formation of quasi-species.

To further evaluate the phenotypic diversity within the viral population, we quantified the diversity of amino acid sequences encoded by the RNA using Shannon entropy. Shannon entropy is a measure

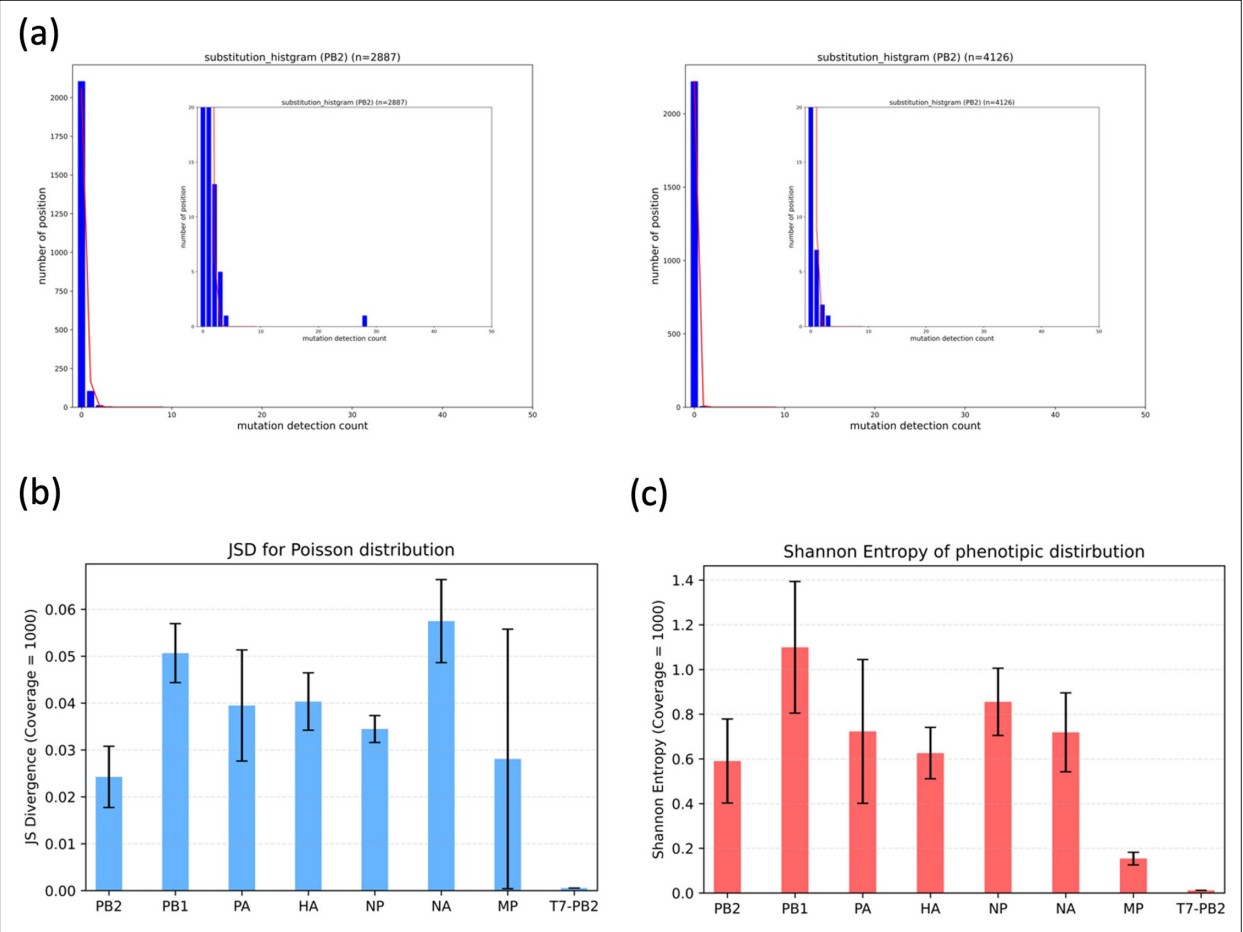

**Figure 4.** The distance from the Poisson distribution and sequence diversity in viral extracted RNA and in vitro transcribed RNA. (**a**) Distribution of mutation detection counts by position in the PB2 gene extracted from Population1 virus (left) and PB2-like sequence RNA synthesized via in vitro transcription (right). The horizontal axis represents the number of times a mutation was detected, while the vertical axis represents the number of positions where mutations with the corresponding detection count were observed. The fitting line represents the Poisson distribution calculated based on the mean value for each dataset. (**b**) Jensen–Shannon diversity between the Poisson distribution and each gene extracted from the virus or the PB2-like RNA synthesized via in vitro transcription. To ensure equal conditions across genes, distributions were normalized to a sequencing coverage of 1000. The NS gene was not included in the graph because no population had sequencing coverage exceeding 1000. The bar height for the viral populations represents the mean value across four populations, and the error bars indicate the standard deviation(n=4). (**c**) Shannon entropy of each gene extracted from the virus and the PB2-like gene synthesized via in vitro transcription. To ensure equal conditions across genes, sequencing results from 1000 molecules were randomly sampled, and the Shannon entropy was calculated for each sample. This procedure was repeated 10 times, and the average value was used as the Shannon entropy for a given gene in a given population. The bar height for the viral populations represents the mean value across four populations, and the error bars indicate the standard deviation(n=4).

of uncertainty based on probability distributions, with higher values indicating greater variability. For each viral gene, the distribution of protein sequences derived from all sequencing fragments (with UMI count ≥3) was constructed, and the entropy values between viral RNA and in vitro transcribed RNA were compared. The results demonstrated that, across the entire gene pool, viral RNA exhibited a higher degree of amino acid sequence diversity in comparison to in vitro transcribed RNA (*Figure 4b*). This increased diversity is likely indicative of the genetic variation accumulated during viral propagation and can serve as an indicator of the potential evolutionary capacity of the virus under the culture conditions utilized in this study.

## Comparison between mutations derived from viral propagation and actual evolutionary lineages

The majority of the mutations measured in this study within viral populations are likely to be neutral mutations that have been randomly introduced. However, it is important to note that each of these

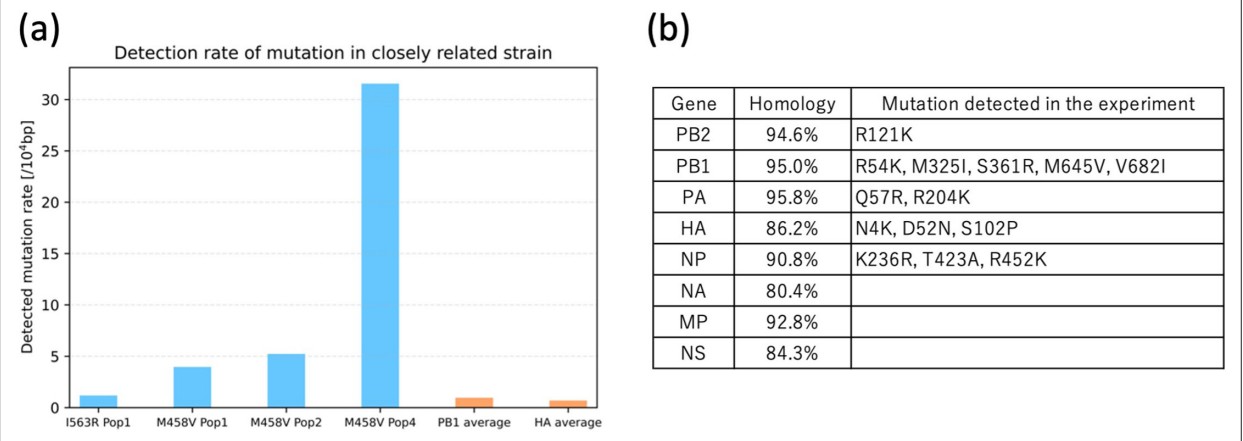

**Figure 5.** Detection rate of variants in related strains and variants matching the latest strain. (**a**) Among the three mutations observed between the strain used in this experiment (PR8) and its related strain (Alaska/1935), the detection rates of the two mutations observed in the cultured population of this experiment within each population are presented. The orange bars represent the average mutation rate in non-homopolymer regions across all populations for PB1 and HA. (**b**) A list of mutations observed in the cultured population of this experiment among the multiple mutations found between PR8 and the latest circulating strain (CA01). 'Homology' indicates the amino acid sequence similarity between PR8 and CA01.

mutations has the potential to become fixed and dominate the population, representing possible origins of emerging variants. A comparison of the viral genome sequences between the strain used in this experiment, A/Puerto Rico/8/1934 (PR8), and its closely related strain, A/Alaska/1935, revealed three amino acid differences: I563R (PB1), D452E (HA), and M458V (HA) (*Qu et al., 2011*; *Zhuang et al., 2019*). Of these, two mutations—I563R (PB1) and M458V (HA)—were detected in at least one of the four experimental populations (*Figure 5a*). These mutations manifested in non-homopolymeric regions, and their frequencies exceeded the mean mutation rate in these regions across all genes, as illustrated in *Figure 3a*. These findings suggest that these mutations may be preferentially introduced during replication or may be positively selected due to their capacity to increase their own copy numbers.

Subsequently, under the assumption that mutations introduced at each nucleotide position follow a Poisson distribution, we identified mutations that occurred at frequencies with a probability of ≤5% as candidates for errors that were positively selected and proliferated. A comparison of these mutations with those found between PR8 and the recent pandemic strain A/California/01/2009 (CA01) revealed that 14 amino acid substitutions were shared (*Smith et al., 2009*). Among the 4450 amino acid residues, 402 differed between PR8 and CA01, and a total of 596 non-synonymous mutations exceeding the threshold were identified in this study. Assuming random selection of positions, the probability that a randomly introduced mutation matches both the position and the resulting amino acid of any of the 402 known substitutions is calculated as follows: the resulting percentage, 1.6%, is then calculated by dividing 402 by 4450 and then multiplying by 0.177. In this context, 0.177 represents the mean probability that two independent random point mutations on the same codon will result in the same amino acid substitution. The z-score for observing 14 coincident mutations is calculated as follows:

$$\lambda = 0.016$$

$$n = 596$$

$$X = 14$$

$$z = \frac{X - n\lambda}{\sqrt{n\lambda}} = 1.43$$

The corresponding one-sided p value is 0.0764. Consequently, no statistically significant difference was observed at the 5% level of significance, indicating that selective mutation introduction driving convergence toward recent pandemic strains cannot be concluded from this dataset.

## Discussion

In this study, we established a methodology for single-molecule genome RNA sequencing within cultured influenza virus populations and demonstrated the potential for experimental mutation prediction through sequence distribution analysis. To validate this approach, the sequencing of UMI-tagged linearized plasmids was conducted. The analysis revealed that focusing solely on sequences with three or more reads that shared the same UMI led to an enhancement in sequence accuracy by more than tenfold. This resulted in an error rate that was on the order of $10^{-5}$ per bp per read. Assuming that PCR and sequencing introduce errors at a rate of approximately $10^{-3}$, the probability of two or more errors occurring at the same base position among three reads theoretically decreases to the order of $10^{-6}$. The discrepancy between this theoretical value and our measured data is likely attributable to mutation-prone regions, such as homopolymers and amplification bias, where error-containing sequences were preferentially amplified and detected as majorities.

It is widely acknowledged that PacBio sequencers frequently generate errors in homopolymer regions, and our study observed analogous trends. Specifically, when the UMI redundancy threshold was set to 3, error rates in homopolymer regions were found to be higher than those in non-homopolymer regions. However, the integration of UMIs has led to a significant reduction in this error gap, thereby validating the efficacy of UMI technology in addressing erroneous reads. This finding indicates that UMI-based methods have the potential to enhance the accuracy of PacBio, especially for genes and sequences that contain homopolymer regions. The implications of this study extend to other single-molecule sequencing platforms such as nanopore.

A subsequent analysis of RNA from virus populations derived from single virus particles exhibited elevated error rates when compared to in vitro transcribed RNA, a phenomenon that is presumably attributable to mutations introduced during viral replication. The observed mutation distribution comprised both mutations consistent with a neutral, Poisson-like accumulation and mutations that deviated substantially from a Poisson distribution. This pattern indicates the coexistence of neutral and non-neutral mutations within the viral population, forming a quasi-species structure. The deviation from Poisson expectations suggests that certain mutations were subject to selective pressures, likely influencing replication efficiency or protein function under the specific culture conditions. For instance, the mutation detection rate near the HA antigenic site (amino acids 180–200) was $1.62 \times 10^{-4}$, approximately 1.5 times higher than the genome-wide mutation rate, highlighting a potential hotspot under positive selection. This finding corroborates prior reports of higher sequence variability in antigenic regions (*Thyagarajan and Bloom, 2014*; *Wu et al., 2020*). On the other hand, the observed mutation rates among genes do not align with the findings from previous phylogenetic research *Eisfeld et al., 2014* on 'highly conserved' and 'highly divergent' genes, suggesting a lack of correlation between the distribution size and the evolvability of each gene. Such discrepancies may reflect differences in observation timing. Traditional phylogenetic analyses capture fixed mutations shaped by long-term selection, while our study detects earlier-stage mutations that have yet to undergo full selective filtering. Thus, the weak correlation with phylogenetic conservation likely arises because many observed mutations are still under selection. A comparison between RNA extracted from viral populations and in vitro transcribed RNA revealed greater protein sequence diversity in the former, as quantified by Shannon entropy. This greater diversity reflects the accumulation of mutations during replication and the latent evolutionary potential of viral populations.

The reference sequences employed for mapping viral genomes in this study were derived from single particles that contributed to the formation of each virus population. Nevertheless, subtle differences were observed among the consensus sequences from four virus populations, suggesting that even within the same PR8 strain, various mutations had accumulated during laboratory passaging, resulting in genetically diverse populations at the outset. This finding suggests that the experimental strain already possessed a mutation pool, and the observed mutation distribution reflects this background diversity. A comprehensive understanding of the effects of long-term passaging on viral population structure and mutation origins is imperative to obtain significant insights.

As this study did not impose specific selective pressures, we did not observe a significant increase of particular mutations previously linked to drug resistance or host adaptation was not observed within the populations. However, resistance mutations such as I38M, which have been demonstrated to confer resistance to the endonuclease inhibitor baloxavir (*Jones et al., 2021*; *Taniguchi et al., 2024*), were detected (see *Figure 3—source data 1 and 2* for a list of all mutations detected). Conversely,

mutations fixed in PR8-related strains were already present in populations derived from single particles. These findings imply that the viral quasi-species may serve as a latent genetic reservoir, from which advantageous variants can be selected in response to environmental pressures. While genetic variation was also detected in HA and NA, we did not impose drug or immune selection pressure in this study. Therefore, we did not expect to observe mutations that are already known to confer major antigenic changes in these proteins, and we consider it difficult to speculate on their functional implications in this context. Nevertheless, the detection of resistance-associated mutations indicates that the quasi-species pool may indeed harbor functionally relevant variation, even in the absence of explicit selective pressures. Thus, the real-time observation of mutation proliferation under diverse culture conditions will yield pivotal insights into the mechanisms underlying existing mutation expansion, thereby facilitating the prediction of novel mutations.

The predominant paradigm in evolutionary biology is the neutral evolution hypothesis, which posits that most evolutionary processes can be explained by random genetic drift. Consequently, elucidating the origins of these evolutionary processes is paramount for making accurate evolutionary predictions. A comprehensive analysis of neutral mutations necessitates the quantification of minor variants. The sUMI method was employed to detect mutations present at 0.1% frequency in populations by sequencing 10,000 molecules. Furthermore, the sequencing error rate was reduced to the order of $10^{-5}$, comparable to reverse transcriptase error rates. This enabled theoretical detection of mutations at a frequency of 0.05% by analyzing over 100,000 molecules with high accuracy. It is anticipated that this approach will yield comprehensive insights into mutation occurrence rates and distributions of mutations in neutral evolution. Furthermore, we have demonstrated the applicability of this method for mutation forecasting by using logistic modeling based on mutation fitness and initial frequencies. Subsequent applications will encompass the comprehensive detection and quantitative estimation of adaptive mutations under diverse environmental conditions, including the presence of drugs and different host species.

In summary, experimental evidence has demonstrated the efficacy of UMI technology in reducing sequencing errors and accurately measuring mutation distributions within viral populations. Furthermore, evidence was presented demonstrating that sequence distributions within individual populations manifest non-random directional biases. With the continued development of methods to quantify mutation bias and latent evolutionary potential, it is anticipated that laboratory-scale prediction of drug-induced mutations and pandemic-capable strains will become a reality. The broad distribution of mutations indicates that viral populations possess diverse mutation pools, where selective pressures enhance robustness through adaptive mutation selection. This mechanism signifies the ability of viruses to adapt to environmental changes with flexibility, thereby providing critical insights for predicting long-term viral evolution and pandemic emergence. A more thorough examination of the roles of mutations in the context of adaptive viral evolution in response to drug treatment is merited.

## Materials and methods
### Cells and virus
MDCK cells, which are derived from canine renal tubular epithelial cells, were obtained from the American Type Culture Collection. The cells were cultivated in Dulbecco's Modified Eagle Medium (FUJIFILM), with the addition of 10% fetal bovine serum (BioWest) and 1% Penicillin–Streptomycin Solution (FUJIFILM). The cells were maintained at 37°C in a humidified atmosphere with 5% $CO_2$. Cell line identity was verified in our laboratory based on the characteristic epithelial morphology and growth properties of MDCK cells (cobblestone-like monolayer formation and stable proliferation). It is also tested for mycoplasma, and only tested negative was used for the experiments.

The A/Puerto Rico/8/1934 (H1N1) strain of influenza A virus was prepared as previously described (*Eisfeld et al., 2014*) and stored at −80°C. The virus stocks were meticulously thawed and subsequently diluted to the appropriate concentrations prior to utilization.

### UMI ligation to control plasmid
The experiment utilized a control plasmid containing an HA-like sequence. The plasmid was subjected to a series of digestions with the restriction enzymes NheI and EcoRI, which were obtained from Nippon Genetics. Following the digestion with NheI, dephosphorylation was carried out. A double-stranded

oligonucleotide containing a UMI and EcoRI-compatible overhangs was ligated to the plasmid using Ligation Mix (Takara Bio).

## In vitro transcription of control RNA

Plasmids containing PB2- and HA-like sequences under the control of a T7 promoter were constructed by RT-PCR and In-Fusion cloning from viral RNA extracts, as they were selected for representing both highly and poorly conserved regions across strains, showing substantial sequence divergence, and having relatively large genome sizes that facilitate read acquisition with Revio. In vitro transcription was performed using the In Vitro Transcription T7 Kit (Takara Bio) with T7 RNA polymerase.

## Single-virus particle infection and RNA extraction

MDCK cells were cultivated to achieve confluency in 6-well plates. Following the removal of the medium and a thorough washing step with phosphate-buffered saline, the cells were infected with the influenza virus at 0.05 PFU/well. The infection was carried out at 35°C for 1 hour with gentle rocking every 15 minutes. After removing the inoculum, the cells were incubated in 2 mL of 1× Minimum Essential Medium at 37°C with 5% $CO_2$. Culture supernatants were collected 3–4 days post-infection from wells where complete CPE was observed. The extraction of viral RNA was conducted using the QIAamp Viral RNA Mini Kit (QIAGEN). Of the 96 wells examined, 4 (4.2%) exhibited signs of successful infection.

## RT-PCR

Reverse transcription was performed using SuperScript IV (Thermo Fisher Scientific), and PCR was carried out using KOD Plus NEO (TOYOBO). The reverse transcription primer consisted of four functional elements arranged from the 5' end to the 3' end: PCR priming site (AGGTTATGCGTTGATACCACTGCTTGC), UMI composed of 15 randomized nucleotides (NNGNNTNNGNNTNNGNNTNNN), population-specific barcode of six nucleotides, and annealing site complementary to the viral genome (AGCRAAAGCAGG). PCR was performed using an average of 4.15×107 input molecules as the template, as quantified by qPCR targeting the HA gene. The cycling conditions were as follows: initial denaturation at 94°C for 2 minutes, followed by 25 cycles of denaturation at 98°C for 10 seconds, and annealing/extension at 68°C for X s (X=80 for PB2, PB1, and PA, 60 for HA, NP, and NA, 40 for MP and NS). All primers were custom synthesized by FASMAC.

## Sequencing

Sequencing using the PacBio Revio platform was outsourced to Takara Bio and Bioengineering Lab.

## Analyzing sequencing results

### Read processing and barcode demultiplexing

HiFi reads with an average quality score ≥60 were first processed using SeqKit (v0.16.1). Reverse-complement sequences were generated and combined with the original reads to count reads whose reverse-complement strand was sequenced.

Barcode sequences were extracted from using seqkit grep. Reads corresponding to the reference and sample barcodes were separated into individual FASTQ files.

### Primer trimming

To remove PCR priming site, cutadapt (*Martin, 2011*) (v4.0) was applied. Reads matching at least 90% identity were retained, and sequences shorter than 20 nt were discarded:

### UMI grouping and variant calling

Consensus-based error correction and variant calling were performed with umierrorcorrect (run_umierrorcorrect.py). Each dataset was aligned to the reference sequence, and consensus sequences for each UMI group were remapped. The pipeline produced BAM alignment files, which were opened with Genetyx-NGS, and variant summary tables were compiled in TSV format.

## Virus particle synthesis and quantification

Virus particle quantification was performed using digital influenza assay (*Tabata et al., 2019*).

## Bioinformatic analysis

Initial demultiplexing of fastq files by population barcode was performed using Seqkit (*Shen et al., 2016*). UMI consensus sequences were generated using UMIerrorcorrect (*Österlund et al., 2022*), and reads were mapped to reference sequences using Genetyx-NGS/Mac. The mapping results were then further processed to extract the base counts at each nucleotide position. The following metrics were calculated:

### JSD

The frequency distribution of sites with k mutations is denoted by $P(k)$. The Poisson distribution with mean count $\lambda$ is denoted by $Q(k)$. The JSD between $P(k)$ and $Q(k)$ was calculated as follows:

$$D_{KL}(P,Q) = \sum_{x} P(x) \log\left(\frac{P(x)}{Q(x)}\right)$$

$$M = \frac{1}{2}(P+Q)$$

$$D_{JS}(P,Q) = \frac{1}{2}\left(D_{KL}(P,M) + D_{KL}(Q,M)\right)$$

### Shannon entropy

The frequency distribution of an amino acid sequence 'x' translated from an RNA sequence obtained by sequencing is denoted by $P(x)$. The Shannon entropy $H$ is calculated as follows:

$$H = -\sum_{i=1}^{n} P(x_i) \log_2 P(x_i)$$

## Growth simulation using a logistic model

Assumed that the time evolution of the particle numbers of wild-type and mutant species, denoted as $N_w$ and $N_m$, respectively, follows a system of differential equations. This assumption was then used to calculate the temporal changes in the number of each particle type.

$$\frac{dN_w}{dt} = r_w N_w \left(1 - \frac{N_w + N_m}{K}\right)$$

$$\frac{dN_m}{dt} = r_m N_m \left(1 - \frac{N_w + N_m}{K}\right)$$

$$K = 10^{13}, r_w = 1$$

## Acknowledgements

This research was supported by the Japan Science and Technology Agency for Core Research for Evolutional Science and Technology, JST CREST, Japan (JPMJCR22N2), and The 2024–2025 UTOPIA AI Research Discovery Program, AMED, Japan (JP223fa627001). KT is supported by the MERIT-WINGS Program at the University of Tokyo.

## Additional information

### Funding

| Funder | Grant reference number | Author |
| --- | --- | --- |
| Japan Science and Technology Agency | 10.52926/jpmjcr22n2 | Kazuhito Tabata |

| Funder | Grant reference number | Author |
| --- | --- | --- |
| Japan Agency for Medical Research and Development | JP223fa627001 | Kazuhito Tabata |
| MERIT-WINGS Program | | Kenji Tamao |

The funders had no role in study design, data collection and interpretation, or the decision to submit the work for publication.

## Author contributions
Kenji Tamao, Data curation, Formal analysis, Investigation, Methodology, Writing – original draft; Hiroyuki Noji, Resources; Kazuhito Tabata, Conceptualization, Resources, Supervision, Funding acquisition, Writing – original draft, Project administration, Writing – review and editing

## Author ORCIDs
Kenji Tamao ⓘ https://orcid.org/0009-0009-0297-6107
Hiroyuki Noji ⓘ https://orcid.org/0000-0002-8842-6836
Kazuhito Tabata ⓘ https://orcid.org/0000-0002-0463-1374

Reviewer #1 (Public review): https://doi.org/10.7554/eLife.108882.3.sa1
Reviewer #2 (Public review): https://doi.org/10.7554/eLife.108882.3.sa2
Author response https://doi.org/10.7554/eLife.108882.3.sa3

# Additional files

## Supplementary files
MDAR checklist

Supplementary file 1. Overview of the sequencing result of each gene.

## Data availability
The raw sequencing data generated in this study have been deposited in the DDBJ Sequence Read Archive under the BioProject accession number PRJDB35851.

The following dataset was generated:

| Author(s) | Year | Dataset title | Dataset URL | Database and Identifier |
| --- | --- | --- | --- | --- |
| Tamao K | 2025 | Heterogeneity of Genetic Sequence within Quasispecies of Influenza Virus Revealed by Single-Molecule Sequencing | https://ddbj.nig.ac.jp/search/entry/bioproject/PRJDB35851 | DDBJ, PRJDB35851 |

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
