## [Editor Report · eLife Assessment]

This study is an **important** contribution to the field of viral sequencing, providing methods for more accurate characterization of viral genetic diversity using long-read sequencing and unique molecular identifiers (UMIs). Although it is a small pilot study, it shows promise as a **convincing**, validated methodology with broad applicability.

---

## [Referee Report · Reviewer #1 (Public review)]

Tamao et al. aimed to quantify the diversity and mutation rate of the influenza (PR8 strain) in order to establish a high-resolution method for studying intra-host viral evolution . To achieve this, the authors combined RNA sequencing with single-molecule unique molecular identifiers (UMIs) to minimize errors introduced during technical processing. They proposed an in vitro infection model with a single viral particle to represent biological genetic diversity, alongside a control model using in vitro transcribed RNA for two viral genes, PB2 and HA.

Through this approach, the authors demonstrated that UMIs reduced technical errors by approximately tenfold. By analyzing four viral populations and comparing them to in vitro transcribed RNA controls, they estimated that ~98.1% of observed mutations originated from viral replication rather than technical artifacts. Their results further showed that most mutations were synonymous and introduced randomly. However, the distribution of mutations suggested selective pressures that favored certain variants. Additionally, comparison with closely related influenza strain (A/Alaska/1935) revealed two positively selected mutations, though these were absent in the strain responsible for the most recent pandemic (CA01).

Overall, the study is well-designed, and the interpretations are strongly supported by the data.

The authors have addressed all the comments from the previous round of reviews. No further concerns.

---

## [Referee Report · Reviewer #2 (Public review)]

Summary:

This manuscript presents a technically oriented application of UMI-based long-read sequencing to study intra-host diversity in influenza virus populations. The authors aim to minimize sequencing artifacts and improve the detection of rare variants, proposing that this approach may inform predictive models of viral evolution. While the methodology appears robust and successfully reduces sequencing error rates, key experimental and analytical details are missing, and the biological insight is modest. The study includes only four samples, with no independent biological replicates or controls, which limits the generalizability of the findings. Claims related to rare variant detection and evolutionary selection are not fully supported by the data presented.

Strengths:

The study addresses an important technical challenge in viral genomics by implementing a UMI-based long-read sequencing approach to reduce amplification and sequencing errors. The methodological focus is well presented, and the work contributes to improving the resolution of low-frequency variant detection in complex viral populations.

Weaknesses:

The application of UMI-based error correction to viral population sequencing has been established in previous studies (e.g., in HIV), and this manuscript does not introduce a substantial methodological or conceptual advance beyond its use in the context of influenza.

The study lacks independent biological replicates or additional viral systems that would strengthen the generalizability of the conclusions. Potential sources of technical error are not explored or explicitly controlled. Key methodological details are missing, including the number of PCR cycles, the input number of molecules, and UMI family size distributions. These are essential to support the claimed sensitivity of the method.

The assertion that variants at {greater than or equal to}0.1% frequency can be reliably detected is based on total read count rather than the number of unique input molecules. Without information on UMI diversity and family sizes, the detection limit cannot be reliably assessed.

Although genetic variation is described, the functional relevance of observed mutations in HA and NA is not addressed or discussed in the context of known antigenic or evolutionary features of influenza. The manuscript is largely focused on technical performance, with limited exploration of the biological implications or mechanistic insights into influenza virus evolution.

The experimental scale is small, with only four viral populations derived from single particles analyzed. This limited sample size restricts the ability to draw broader conclusions about quasispecies dynamics or evolutionary pressures.

Comments on revisions:

The revised manuscript provides additional methodological detail and clearer presentation, which improves transparency. However, the main limitations persist: the study remains small in scale, lacks independent validation, and relies on theoretical rather than empirical support for its claimed detection sensitivity. As a result, the work represents a modest technical advance rather than a substantive contribution to understanding influenza virus evolution.

---

## [Author Response]

The following is the authors’ response to the original reviews.

**Reviewer #1 (Public review):**
(1) The methods section is overly brief. Even if techniques are cited, more experimental details should be included. For example, since the study focuses heavily on methodology, details such as the number of PCR cycles in RT-PCR or the rationale for choosing HA and PB2 as representative in vitro transcripts should be provided.

We thank the reviewer for this important suggestion. We have now expanded the Methods section to include the number of PCR cycles used in RT-PCR (line 407) and have explained the rationale for choosing HA and PB2 as representative transcripts (line 388).

(2) Information on library preparation and sequencing metrics should be included. For example, the total number of reads, any filtering steps, and quality score distributions/cutoff for the analyzed reads.

We agree and have added detailed information on library preparation, filtering criteria, quality score thresholds, and sequencing statistics for each sample (line 422, Figure S2).

(3) In the Results section (line 115, "Quantification of error rate caused by RT"), the mutation rate attributed to viral replication is calculated. However, in line 138, it is unclear whether the reported value reflects PB2, HA, or both, and whether the comparison is based on the error rate of the same viral RNA or the mean of multiple values (as shown in Figure 3A). Please clarify whether this number applies universally to all influenza RNAs or provide the observed range.

We appreciate this point. We have clarified in the Results (line 140) that the reported value corresponds to PB2.

(4) Since the T7 polymerase introduced errors are only applied to the in vitro transcription control, how were these accounted for when comparing mutation rates between transcribed RNA and cell-culture-derived virus?

We agree that errors introduced by T7 RNA polymerase are present only in the in vitro–transcribed RNA control. However, even when taking this into account, the error rate detected in the in vitro transcripts remained substantially lower than that observed in the viral RNA extracted from replicated virus (line 140, Fig.3a). Thus, the difference cannot be explained by T7-derived errors, and our conclusion regarding the elevated mutation rate in cell-culture–derived viral populations remains valid.

(5) Figure 2 shows that a UMI group size of 4 has an error rate of zero, but this group size is not mentioned in the text. Please clarify.

We have revised the Results (line 98) to describe the UMI group size of 4.

**Reviewer #2 (Public review):**
(1) The application of UMI-based error correction to viral population sequencing has been established in previous studies (e.g., HIV), and this manuscript does not introduce a substantial methodological or conceptual advance beyond its use in the context of influenza.

We appreciate the reviewer’s comment and agree that UMI-based error correction has been applied previously to viral population sequencing, including HIV. However, to our knowledge, relatively few studies have quantitatively evaluated both the performance of this method and the resulting within-quasi-species mutation distributions in detail. In our manuscript, we not only validate the accuracy of UMIbased error correction in the context of influenza virus sequencing, but also quantitatively characterize the features of intra-quasi-species distributions, which provides new insights into the mutational landscape and evolutionary dynamics specific to influenza. We therefore believe that our work goes beyond a simple application of an established method.

(2) The study lacks independent biological replicates or additional viral systems that would strengthen the generalizability of the conclusions.

We agree with the reviewer that the lack of independent biological replicates and additional viral systems limits the generalizability of our findings. In this study, we intentionally focused on single-particle–derived populations of influenza virus to establish a proof-of-principle for our sequencing and analytical framework. While this design provided a clear demonstration of the method’s ability to capture mutation distributions at the single-particle level, we acknowledge that additional biological replicates and testing across diverse viral systems would be necessary to confirm the broader applicability of our observations. Importantly, even within this limited framework, our analysis enabled us to draw conclusions at the level of individual viral populations and to suggest the possibility of comparing their mutation distributions with known evolvability. This highlights the potential of our approach to bridge observations from single particles with broader patterns of viral evolution. In future work, we plan to expand the number of populations analyzed and include additional viral systems, which will allow us to more rigorously assess reproducibility and to establish systematic links between mutation accumulation at the single-particle level and evolutionary dynamics across viruses.

(3) Potential sources of technical error are not explored or explicitly controlled. Key methodological details are missing, including the number of PCR cycles, the input number of molecules, and UMI family size distributions.

We thank the reviewer for this important suggestion. We have now expanded the Methods section to include the number of PCR cycles used in RT-PCR (line 407). In addition, we have added information on the estimated number of input molecules. Regarding the UMI family size distributions, we have added the data as Figure S2 and referred to it in the revised manuscript.

Finally, with respect to potential sources of technical error, we note that this point is already addressed in the manuscript by direct comparison with in vitro transcribed RNA controls, which encompass errors introduced throughout the entire experimental process. This comparison demonstrates that the error-correction strategy employed here effectively reduces the impact of PCR or sequencing artifacts.

(4) The assertion that variants at ≥0.1% frequency can be reliably detected is based on total read count rather than the number of unique input molecules. Without information on UMI diversity and family sizes, the detection limit cannot be reliably assessed.

We thank the reviewer for raising this important issue. We agree that our original description was misleading, as the reliable detection limit should not be defined solely by total read count. In the revised version, we have added information on UMI distribution and family sizes (Figure S2), and we now state the detection limit in terms of consensus reads. Specifically, we define that variants can be reliably detected when ≥10,000 consensus reads are obtained with a group size of ≥3 (line 173).

(5) Although genetic variation is described, the functional relevance of observed mutations in HA and NA is not addressed or discussed.

We appreciate the reviewer’s suggestion. In our study, we did not apply drug or immune selection pressure; therefore, we did not expect to detect mutations that are already known to cause major antigenic changes in HA or NA, and we think it is difficult to discuss such functional implications in this context. However, as noted in discussion, we did identify drug resistance–associated mutations. This observation suggests that the quasi-species pool may provide functional variation, including resistance, even in the absence of explicit selective pressure. We have clarified this point in the text to better address the reviewer’s concern (line 330).

(6) The experimental scale is small, with only four viral populations derived from single particles analyzed. This limited sample size restricts the ability to draw broader conclusions.

We thank the reviewer for pointing out the limitation of analyzing only four viral populations derived from single particles. We fully acknowledge that the small sample size restricts the generalizability of our conclusions. Nevertheless, we would like to emphasize that even within this limited dataset, our results consistently revealed a slight but reproducible deviation of the mutation distribution from the Poisson expectation, as well as a weak correlation with inter-strain conservation. These recurring patterns highlight the robustness of our observations despite the sample size.

In future work, we plan to expand the number of viral populations analyzed and to monitor mutation distributions during serial passage under defined selective pressures. We believe that such expanded analyses will enable us to more reliably assess how mutations accumulate and to develop predictive frameworks for viral evolution.

**Reviewer #1 (Recommendations for the authors):**
(1) Please mention Figure 1 and S2 in the text.

Done. We now explicitly reference Figures 1 and S2 (renamed to S1 according to appearance order) in the appropriate sections (lines 74, 124).

(2) In Figure 4A, please specify which graph corresponds to PB2 and which to PB2-like sequences.

Corrected. Figure 4A legend now specify PB2 vs. PB2-like sequences.

(3) Consider reducing redundancy in lines 74, 149, 170, 214, and 215.

We thank the reviewer for this stylistic suggestion. We have revised the text to reduce redundancy in these lines.

**Reviewer #2 (Recommendations for the authors):**
(1) The manuscript states that "with 10,000 sequencing reads per gene ...variants at ≥0.1% frequency can be reliably detected." However, this interpretation conflates raw read counts with independent input molecules.

We have revised this statement throughout the text to clarify that sensitivity depends on the number of unique UMIs rather than raw read counts (line 173). To support this, we calculated the probability of detecting a true variant present at a frequency of 0.1% within a population. When sequencing ≥10,000 unique molecules, such a variant would be observed at least twice with a probability of approximately 99.95%. In contrast, the error rate of in vitro–transcribed RNA, reflecting errors introduced during the experimental process, was estimated to be on the order of 10⁻⁶ (line 140, Fig. 3a). Under this condition, the probability that the same artificial error would arise independently at the same position in two out of 10,000 molecules is <0.5%. Therefore, variants present at ≥0.1% can be reliably distinguished from technical artifacts and are confidently detected under our sequencing conditions.

(2) To support the claimed sensitivity, please provide for each gene and population: (a) UMI family size distributions, (b) number of PCR cycles and input molecule counts, and (c) recalculation of the detection limit based on unique molecules.If possible, I encourage experimental validation of sensitivity claims, such as spike-in controls at known variant frequencies, dilution series, or technical replicates to demonstrate reproducibility at the 0.1% detection level.

We have added (a) histograms of UMI family size distributions for each gene and population (Figure S2), (b) detailed method RT-PCR protocol and estimated input counts (line 407), and (c) recalculated detection limits (line 173).

We appreciate the reviewer’s suggestion and fully recognize the value of spike-in experiments. However, given the observed mutation rate of T7-derived RNA and the sufficient sequencing depth in our dataset, it is evident that variants above the 0.1% threshold can be robustly detected without additional spike-in controls.